# Patients' experiences with the application of medical adhesives to the skin: a qualitative systematic review protocol

Hannelore Hofman [1] Dimitri Beeckman [1,2] Tanja Duljic [2,3]
Samal Al Gilani,[4] Sara Johansson,[5] Jan Kottner [1,6] Lise-Marie Kinnaer [1]
Mats Eriksson [7]

For numbered affiliations see end of article.

**Correspondence to**
Professor Dr Mats Eriksson;
mats.h.eriksson@oru.se

## ABSTRACT

**Introduction** Medical adhesives are adhesives used in medical devices to establish and maintain contact with the body over a period of time (usually by application to the skin) and are widely used in most care settings. Application of medical adhesives to the skin can lead to skin stripping, mild or severe allergic reactions and skin irritation that may manifest as redness, itching or rash. Adhesive-related skin injury can lead to infection, delayed wound healing and an increased risk of scarring. These injuries can cause severe discomfort and pain, and can affect the patient's quality of life. A systematic review summarising patient's experiences on this topic will contribute to informing adhesive producers and policy makers, and guiding further development and improvement of available technologies.

**Methods and analysis** This systematic review protocol is based on the principles of the Preferred Reporting Items for Systematic Review and Meta-Analysis Protocols guideline. A systematic search will be conducted in CINAHL, EMBASE, MEDLINE and PsycINFO. In addition, manual searches will be performed, reviewing the reference lists of relevant reviews and articles included for quality assessment. Qualitative studies using various methods will be considered for inclusion. Screening of title, abstract and full text will be done by two reviewers. The methodological quality of studies under consideration will be critically assessed by two reviewers using the Joanna Briggs Institute Critical Appraisal Tool for Qualitative Research. Data extraction will be performed independently by two reviewers using a predefined data extraction form. Meta-aggregation will be used to summarise the evidence.

**Ethics and dissemination** No ethical approval or consent is required because no participants will be recruited. This systematic review protocol is published in an open access journal to increase transparency of the research methods used. Results will be disseminated at national and international conferences.

## INTRODUCTION
### Rationale

Medical adhesives provide securement for medical devices, facilitate skin protection and healing, and allow non-invasive monitoring.[1]

### STRENGTHS AND LIMITATIONS OF THIS STUDY

⇒ The results of this systematic review will be reported following the Enhancing Transparency in Reporting the Synthesis of Qualitative Research (ENTREQ) statement.
⇒ The literature screening, quality assessment and data extraction will be performed by two reviewers to minimise bias.
⇒ The search strategy is developed in collaboration with an expert library technician and adapted to four electronic databases, relevant to this field.
⇒ This systematic review will only include articles published in English, Swedish, Dutch, German, Norwegian or Danish, which may introduce language bias.

Several definitions for medical adhesives are used in the literature.[1–3] In this study, medical adhesives are defined as adhesives used in medical devices to establish and maintain contact with the body over a period of time (usually by application to the skin). They are a component of a variety of healthcare products, including tapes, dressings, electrodes, ostomy supplies and patches.[1]

Medical adhesives are used in all patients groups in most care settings, billions of times every year.[4 5] In a cardiac-telemetry unit and a medical-surgical unit in an acute care facility in the Midwestern United States, a median number of 6.25 and 3.00 adhesive products were used per patient per day, respectively. Electrodes, peripheral intravenous dressings, tape, surgical dressings, surgical closures, wound dressings and peripherally inserted central catheter dressings were the most frequently used adhesive products. The number of medical adhesive products used per patient, varied from 0 to 24.[6]

Application of medical adhesives to the skin can lead to skin stripping, mild or severe

allergic reactions and skin irritation that may manifest as redness, itching or rash.[1] Skin stripping injury related to the use of medical adhesives occurs when the attachment between the skin and a medical adhesive is stronger than that between the individual skin cells. This causes the epidermal layers to separate or the epidermis to detach from the dermis.[1] Adhesive-related skin stripping injuries can occur at any age and in any clinical setting but are especially prevalent in the elderly and neonates, who may have fragile skin.[7–9] This can lead to inflammatory skin reactions, oedema and soreness, all of which can have an adverse effect on the skin barrier function.[7 10 11] These injuries can cause severe discomfort and pain, and can affect the patient's quality of life.[12–15] The International Association for the Study of Pain defines pain as 'an unpleasant sensory and emotional experience associated with, or resembling that associated with, actual or potential tissue damage'.[16]

Some studies and guidelines on the prevention of medical adhesive-related skin injuries (MARSI) mention the risk of pain but few authors focus on patient's experiences such as pain, anxiety and discomfort.[12 17–19] The most recent review on this topic was not conducted in a systematic manner and focused on pain caused by repeated tape and dressing removal.[13] Delayed wound healing due to pain, the impact on the patient's quality of life and awareness of healthcare practitioners were explored. However, the focus was limited to tapes and wound dressings, and did not take other types of medical adhesives into account. In addition, continuous progress is being made in the development of adhesives. Current medical adhesives adhere too strongly to the skin, resulting in MARSI during removal or presence on the skin.[7 20] Consequently, adhesives with sensitive removability are being developed.[4 21 22] Due to these recent advances in adhesive research, this systematic review aims to summarise patients' experiences with the application of medical adhesives to the skin. This review will contribute to informing adhesive producers and policy makers, and guiding further development and improvement of available technologies.

## Research question

What are the experiences of patients with the application of medical adhesives to the skin?

## METHODS

This systematic review protocol is based on the principles of the Preferred Reporting Items for Systematic Review and Meta-Analysis Protocols (PRISMA-P) criteria.[23] Meta-aggregation will be used to synthesise the results. Meta-aggregation is an approach that takes into account the nature and traditions of qualitative research, while also following the rigorous process of systematic review.[24] This method of synthesis aims to enable generalisable recommendations to guide practitioners and policy makers[25] by developing knowledge in an unbiased way, not influenced by the reviewer or outside factors.[26]

## Eligibility criteria

### Population and context

The focus of this review will be on patients who currently or in the past have had medical adhesives applied to the skin. For the purpose of this review, there will be no restrictions on the age or sex of the population studied.

### Phenomena of interest

The phenomenon of interest in this review will be the experience of patients with the application of medical adhesives to the skin.

### Study design

Qualitative studies using various methods will be eligible. Qualitative studies and qualitative data from mixed method studies that describe the experience of patients with the application of medical adhesives to the skin will be considered.[27]

### Setting, time frame and language

There will be no restrictions on the setting. Articles published between January 2012 and November 2022 will be considered. The search period will be restricted to the last 10 years because new medical adhesives are being developed and technological advances are being made continuously.[4 20 22] As a result, this review will focus on medical adhesives that currently are being used in clinical practice. Articles published in languages in which at least two members of the review team are proficient, namely English, Swedish, Dutch, German, Danish and Norwegian, will be included in this systematic review.

## Search strategy and information sources

To identify relevant studies, a two-step strategy will be used. First, a systematic search will be conducted in the following electronic databases: CINAHL (accessed through the EBSCO interface), EMBASE (accessed through Elsevier), MEDLINE (accessed through the Ovid interface) and PsycINFO (accessed through the EBSCO interface). Concepts used for the initial searches in MEDLINE (accessed through the Ovid interface) will be experience (keywords include 'pain', 'dermatitis', 'itching', 'pruritus' and 'discomfort') and removal of dressings (keywords include 'adhesive', 'bandage', 'dressing', 'adverse event', 'device deficiency', 'removal', 'change' and 'application'). The initial search strategy, including all identified keywords and index terms, will be customised for each electronic database (see online supplemental file). Second, a manual search will be performed, reviewing the reference lists of relevant review articles and articles included for quality assessment to identify studies not covered by the search strategy.

## Study selection, data collection and management

All databases will be searched individually. The results of the database searches will be exported to Covidence

software for systematic reviews, which allows the recording of reasons for exclusion. Duplicates will be tracked down and subsequently removed. Literature screening will be performed by two independent reviewers. Screening will consist of two steps: (a) screening of titles and abstracts against the inclusion criteria and (b) screening of the selected full-text articles. In case of disagreement, discussions will be held until consensus is reached. The search and selection process will be summarised in a PRISMA flowchart.

### Assessment of methodological quality

The methodological quality of the qualitative studies under consideration will be critically assessed by two reviewers using the Joanna Briggs Institute Critical Appraisal Tool for Qualitative Research. In case of disagreement between the two reviewers, a decision will be made by consensus. If necessary, a third reviewer will be consulted.

### Data extraction

Prior to database search, a data extraction form will be prepared that includes the following categories: (a) bibliographic information (lead author, year, title, journal, full citation), (b) study design and sample size, (c) patient demographics, medical history, setting and geographical context, (d) description of how the research findings are addressed in the article, (e) method of data collection, (f) method of data analysis, (g) context (product names/brands or type of material of medical adhesives investigated), (h) phenomenon of interest (experience of patients with the application of medical adhesives to the skin) and (i) findings. A finding in meta-aggregation is defined as 'a verbatim extract of the author's analytical interpretation of the results or data'.[27] Each extracted finding is to be accompanied by a direct quotation of the participant voice (ie, an illustration) from the same study.[27 28] Authors of included articles will be contacted if further clarifications concerning the conducted research are needed.

Data extraction will be performed independently by two reviewers. Any ambiguities will be discussed within the research team. Final data extraction will be done through discussions based on each data extraction form until consensus is reached. Quality control of the extracted data will be performed by another member of the research team on 20% of the included articles.

### Data synthesis

It is expected that the search query of this study will result in a large number of studies, some of which are of poor methodological quality. Meta-aggregation will be used to summarise the evidence. To each finding, the reviewer will allocate a level of plausibility. Three levels of plausibility exist: unequivocal (ie, findings accompanied by an illustration that is beyond reasonable doubt), equivocal (ie, findings accompanied by an illustration lacking clear association with it and therefore open to challenge) and unsupported (ie, findings that are not supported by the data). Unsupported findings will not appear in the data synthesis.[27 28] In meta-aggregation, the following three steps will be conducted: (a) all findings will be extracted from the results, discussion and conclusion section of all included studies, accompanied by an illustration and will be allocated a level of plausibility (ie, unequivocal, equivocal or unsupported), (b) findings will be summarised into categories (ie, a brief description of a key concept arising from the aggregation of two or more like findings) based on similarity in meaning and (c) synthesised findings (ie, an overarching description of a group of categorised findings) will be derived from categories.[27 28]

The extracted findings will be presented in the text, both as a narrative summary and in a matrix consistent with the aim of this review. The report will be structured following the Enhancing Transparency in Reporting the Synthesis of Qualitative Research (ENTREQ) statement.[29] The results will be synthesised and conceptualised at a higher level of abstraction to draw conclusions.[27 28]

The relationships between the characteristics of each study and the phenomena of interest they report, as well as the relationships between the phenomena of interest of different studies, will be examined narratively by comparing and contrasting these relationships across studies. Attention will be paid to potential differences between studies, including methodological differences. Differences in patient's experiences in various age groups will be examined by comparing and contrasting findings between studies for the following age groups: newborns (0–1 month), children (2 months to 11 years), adolescents (12–18 years), adults (19–74 years) and elderly (75 years and older).

### Patient and public involvement

No patients will be involved in the design or conduct of this review.

### ETHICS AND DISSEMINATION

This review will use published literature and will not recruit participants. Therefore, no formal ethical approval or consent is necessary. This systematic review protocol is published in an open access journal to increase transparency of the research methods used.

This systematic review will include studies that have received formal ethical approval. This systematic review will likely provide a detailed summary of the experiences of patients with the application of medical adhesives to the skin. This review might also provide recommendations on strategies to improve patient's experience with the application of medical adhesives to the skin.

The results of this systematic review will be published in a leading peer-reviewed journal in the field. Results will also be disseminated at national and international conferences.

**Author affiliations**
¹University Centre for Nursing and Midwifery, Department of Public Health and Primary Care, Faculty of Medicine and Health Sciences, Ghent University, Ghent, Belgium
²Swedish Centre for Skin and Wound Research (SCENTR), Faculty of Medicine and Health, School of Health Sciences, Örebro University, Örebro, Sweden
³Department of Care Science, Malmö University, Malmö, Sweden
⁴School of Health and Welfare, Dalarna University, Falun, Sweden
⁵Creative Mammals, Gothenburg, Sweden
⁶Institute of Clinical Nursing Science, Charité Center for Health and Human Sciences, Charité Universitätsmedizin, Berlin, Germany
⁷Faculty of Medicine and Health, School of Health Sciences, Örebro University, Örebro, Sweden

**Acknowledgements** Mölnlycke Health Care AB, award/grant number: NA.

**Contributors** All authors contributed to the conception of the research question and the writing of the protocol. HH, DB, SJ, JK, LMK and ME contributed to the development of search strategies, eligibility criteria and methodology for data synthesis. All authors contributed to the draft protocol and approved the final version of this protocol. HH, TD, SAG and ME will work in duplicate to review the titles and abstracts of all materials obtained using the search strategy to exclude articles that do not meet the eligibility criteria. HH, TD, SAG and ME will evaluate potentially eligible studies through full-text screening and exclude non-eligible studies, documenting the reason for exclusion. HH, TD, SAG and ME will independently extract data from the included studies. HH, TD, SAG and ME will analyse the data and draft the manuscript. All authors will read, provide feedback and approve the final manuscript.

**Funding** Mölnlycke Health Care AB, award/grant number: NA. Mölnlycke Health Care AB is not involved in any other aspect than the funding of this systematic review. The funder will have no input on the interpretation or publication of the study results.

**Competing interests** None declared.

**Patient and public involvement** Patients and/or the public were not involved in the design, or conduct, or reporting, or dissemination plans of this research.

**Patient consent for publication** Not applicable.

**Provenance and peer review** Not commissioned; externally peer reviewed.

**ORCID iDs**
Hannelore Hofman http://orcid.org/0000-0001-5750-0346
Dimitri Beeckman http://orcid.org/0000-0003-3080-8716
Tanja Duljic http://orcid.org/0009-0007-8496-5609
Jan Kottner http://orcid.org/0000-0003-0750-3818
Lise-Marie Kinnaer http://orcid.org/0000-0003-3666-8128
Mats Eriksson http://orcid.org/0000-0002-5996-2584

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
