## [Reviewer comments · BMJ Open]

ARTICLE DETAILS

TITLE (PROVISIONAL)	Patients' experiences with the application of medical adhesives to the skin: a qualitative systematic review protocol
AUTHORS	Hofman, Hannelore; Beeckman, Dimitri; Duljic, Tanja; Al Gilani, Samal; Johansson, Sara; Kottner, Jan; Kinnaer, Lise-Marie; Eriksson, Mats

VERSION 1 – REVIEW

REVIEWER	Abu-Qamar , Ma'en Zaid Edith Cowan University
REVIEW RETURNED	30-Mar-2023

GENERAL COMMENTS	I got a chance to review the article entitled "Patients' experiences with the application of medical adhesives to the skin: a protocol for a systematic review and meta-aggregation." – for BMJ Open. This is a protocol of an interesting topic. However, I suggest reviewing the protocol in the light of the following comments. The title is wordy. Superfluous words can be removed, and so the title can be reworded in a better way "Medical adhesives: a protocol of a meta-aggregation of patients' experiences with skin applications" OR "Medical adhesives: a protocol of a systematic review of patients' experiences with skin applications" OR "Medical adhesives: a protocol of a meta-synthesis of patients' experiences with skin applications" The project/ review takes a qualitative perspective, the term "empirical evidence" is often used in a quantitative context. Abstract > methods and analysis> elaborate on the process of meta-aggregation The rationale looks reasonable. However, areas need to be supported with references. Examples of these include page 4, line 21 – line 24 Mixed method studies might contain relevant qualitative data. Will you be excluding mixed method studies? Your answer needs to be justified. It is stated that publications will be included if published from 2012 onwards. I am wondering of the reason for selecting this timeframe. Will you be contacting authors of included articles for further clarifications or to include unpublished research? I cannot see that data extraction will include information related to patient demography and health history. These factors might have
---

	effects on the person experience. Page 6, data extraction, line 58, can you please outline the basis on which this assumption was made. Information provided about meta-aggregation is very brief, and it is backed up with outdated references. There are more updated references discussing meta-aggregation. I provided the citation of one of them. Lockwood C, Porritt K, Munn Z, Rittenmeyer L, Salmond S, Bjerrum M, Loveday H, Carrier J, Stannard D. Chapter 2: Systematic reviews of qualitative evidence. In: Aromataris E, Munn Z (Editors). JBI Manual for Evidence Synthesis. JBI, 2020. Available from https://synthesismanual.jbi.global. https://doi.org/10.46658/JBIMES-20-03 Information provided on ethics is reasonable. However, it is worth declaring your expectations of the included studies in terms of ethics. Most methodological references are outdated. There are more updated references that can be cited. I provided an example above.
--	---

REVIEWER	Kaul, Subuhi John H Stroger Jr Hospital of Cook County
REVIEW RETURNED	18-Apr-2023

GENERAL COMMENTS	Thank you for your work. This is a comprehensive study protocol. I have one suggestion, the authors could consider adding contact dermatitis as a side effect of adhesive use and hence expand their search to include "itch", "pruritus", "dermatitis" and "rash". This is a common side effect of adhesive use as well.
---

VERSION 1 – AUTHOR RESPONSE

Reviewer: 1
Dr. Ma'en Zaid Abu-Qamar , Edith Cowan University

Comments to the Author:

Dear authors,

I got a chance to review the article entitled "Patients' experiences with the application of medical adhesives to the skin: a protocol for a systematic review and meta-aggregation." – for BMJ Open. This is a protocol of an interesting topic. However, I suggest reviewing the protocol in the light of the following comments.

The title is wordy. Superfluous words can be removed, and so the title can be reworded in a better way "Medical adhesives: a protocol of a meta-aggregation of patients' experiences with skin applications" OR "Medical adhesives: a protocol of a systematic review of patients' experiences with skin applications" OR "Medical adhesives: a protocol of a meta-synthesis of patients' experiences with skin applications"

Response: Thank you for your suggestion. However, when revisiting the author guidelines of BMJ Open, we believe that the title fits the author guidelines and gives a concise and informative overview of the topic of this manuscript.

The project/ review takes a qualitative perspective, the term "empirical evidence" is often used in a quantitative context.

Response: The wording in the manuscript (line 208) has been updated. The word 'empirical' has been

removed from the sentence. The abstract has been updated accordingly (line 55).

Abstract > methods and analysis> elaborate on the process of meta-aggregation

Response: An updated full description of the method used to synthesize the qualitative data, namely meta-aggregation, is provided in the full text (line 206-225). Due to only a limited number of words being allowed in the abstract, it is not possible to elaborate this process without removing other important information on the methodology used. Therefore, no changes were made to the abstract in the light of this suggestion.

The rationale looks reasonable. However, areas need to be supported with references. Examples of these include page 4, line 21 – line 24

Response: Upon revisiting the rationale, every statement made in the rationale is already supported by at least one reference. In some cases, successive statements are supported by the same reference. In those cases, like the area you pointed out (in this manuscript version: line 86-91), the article in question is referenced at the end of the statements (multiple sentences) that are supported by this article.

Mixed method studies might contain relevant qualitative data. Will you be excluding mixed method studies? Your answer needs to be justified.

Response: Every study that provides further insights in the experience of patients with the application of medical adhesives to the skin (using qualitative data) will be considered for eligibility. Qualitative data from mixed methods studies that meet the inclusion criteria will therefore also be considered in this systematic review. This was added to the manuscript (line 144-146)

It is stated that publications will be included if published from 2012 onwards. I am wondering of the reason for selecting this timeframe.

Response: New medical adhesives are being developed and technological advances are being made continuously. In order to be able to focus on medical adhesives that currently are being used in clinical practice, we decided to include evidence covering the last 10 years. We added this explanation to the text (line 150-153).

Will you be contacting authors of included articles for further clarifications or to include unpublished research?

Response: Authors of included articles will be contacted if further clarifications concerning the conducted research are needed. This was added to the manuscript on line 196-198.

I cannot see that data extraction will include information related to patient demography and health history. These factors might have effects on the person experience.

Response: Thank you for your suggestion. The authors have replaced 'population' with 'patient demographics' and will add 'medical history' to the data extraction table. The manuscript has been updated accordingly (line 189).

Page 6, data extraction, line 58, can you please outline the basis on which this assumption was made.

Response: This assumption was made based on the preliminary searches that were performed to finetune the search query. When entering the search query in the MEDLINE database (accessed through Ovid), over 1700 results were retrieved. Because of this and the focus of this study and its search query being on patient experiences, it was assumed that a large number of qualitative studies, some of which of poor methodological quality, would be retrieved.

Information provided about meta-aggregation is very brief, and it is backed up with outdated references. There are more updated references discussing meta-aggregation. I provided the citation of one of them.

Lockwood C, Porritt K, Munn Z, Rittenmeyer L, Salmond S, Bjerrum M, Loveday H, Carrier J, Stannard D. Chapter 2: Systematic reviews of qualitative evidence. In: Aromataris E, Munn Z (Editors). JBI Manual for Evidence Synthesis. JBI, 2020. Available from [https://eur03.safelinks.protection.outlook.com/?url=https%3A%2F%2Fdoi.org%2F10.46658%2FJBIMES-20-03&data=05%7C01%7CHannelore.Hofman%40ugent.be%7C40bb0de0fc3140b647f408db4008fbfe%](https://eur03.safelinks.protection.outlook.com/?url=https%3A%2F%2Fdoi.org%2F10.46658%2FJBIMES-20-03&data=05%7C01%7CHannelore.Hofman%40ugent.be%7C40bb0de0fc3140b647f408db4008fbfe%2F)

7Cd7811cdeecef496c8f91a1786241b99c%7C1%7C0%7C638174179549242814%7CUnknown%7CTWFpbGZsb3d8eyJWIjoiMC4wLjAwMDAiLCJQIjoiV2luMzliLCJBTil6Ik1haWwiLCJXVCi6Mn0%3D%7C3000%7C%7C&sdata=FR1TUm3QYV0gpoNqVYvQiKBYPd5MNvzXf0g1P0YrpNY%3D&reserved=0

Response: More recent references and additional elaboration on definitions used in meta-aggregation have been added to the manuscript (line 193-196; line 206-220; line 224-225). The reference list has been updated accordingly.

Information provided on ethics is reasonable. However, it is worth declaring your expectations of the included studies in terms of ethics.

Response: The paragraph on 'Ethics and dissemination' (line 242) has been updated based on this suggestion.

Most methodological references are outdated. There are more updated references that can be cited. I provided an example above.

Response: More recent references and additional elaboration on definitions used in meta-aggregation have been added to the protocol manuscript (line 193-196; line 206-220; line 224-225). The reference list has been updated accordingly.

Reviewer: 2

Dr. Subuhi Kaul, John H Stroger Jr Hospital of Cook County

Comments to the Author:

Thank you for your work. This is a comprehensive study protocol. I have one suggestion, the authors could consider adding contact dermatitis as a side effect of adhesive use and hence expand their search to include "itch", "pruritus", "dermatitis" and "rash". This is a common side effect of adhesive use as well.

Response: Thank you for your suggestion. We are aware that contact dermatitis is a common adverse event that occurs when applying medical adhesives to the skin. Upon developing the initial search query, the search terms "pruritus", "dermatitis" and "itching" were taken into account. These search terms were added to the manuscript on line 161-165. However, because adding these search terms did not result in many additional hits, we decided to remove these search terms from the query in agreement with experienced university library technicians.

Once again, we would like to thank the editor and reviewers for their insightful comments and suggestions. We look forward to hearing back from you soon.

VERSION 2 – REVIEW

REVIEWER	Abu-Qamar , Ma'en Zaid Edith Cowan University
REVIEW RETURNED	20-May-2023

GENERAL COMMENTS	The revision has improved the quality of the protocol. However, I suggest reviewing the protocol in the light of the following suggestion. The title:- The title contains the words "a systematic review and meta-aggregation". It will be interesting to outline the reason for including the two words in the title. Given that the review will employ Joanna Briggs Institute (JBI) approach of meta-aggregation, it will be better to consider the JBI suggestion regarding the title of meta-aggregation. I am wondering of reason for citing the JBI manual for reviewers>
--

	2014 version; whereas the JBI has published updated version. The last version was published in 2020. Given that the review will include publications from a range of languages, it will be necessary to outline how data in different language will be handled to main rigour. The justification is mainly based on outdated reference. Out 27 sources cited in the introduction, 10 were published in the year 2010 or prior.
--	---

VERSION 2 – AUTHOR RESPONSE

Reviewer: 1

Dr. Ma'en Zaid Abu-Qamar, Edith Cowan University

Comments to the Author:

The revision has improved the quality of the protocol. However, I suggest reviewing the protocol in the light of the following suggestion.

The title:-

The title contains the words “a systematic review and meta-aggregation”. It will be interesting to outline the reason for including the two words in the title. Given that the review will employ Joanna Briggs Institute (JBI) approach of meta-aggregation, it will be better to consider the JBI suggestion regarding the title of meta-aggregation.

Response: Thank you for this suggestion. The title has been changed in light of the title suggestion in the JBI manual for evidence synthesis (2020 edition). The title in the manuscript and the ScholarOne Manuscripts portal is now as follows: “Patients’ experiences with the application of medical adhesives to the skin: a qualitative systematic review protocol”.

I am wondering of reason for citing the JBI manual for reviewers> 2014 version; whereas the JBI has published updated version. The last version was published in 2020.

Response: Thank you for bringing this to our attention. The reference to the 2014 edition of the JBI Reviewers’ Manual has been removed from the manuscript (line 148; line 198). The reference list has been updated accordingly.

Given that the review will include publications from a range of languages, it will be necessary to outline how data in different language will be handled to main rigour.

Response: A clarification on how we will handle articles published in different languages, has been added to the manuscript (line 155-157).

The justification is mainly based on outdated reference. Out 27 sources cited in the introduction, 10 were published in the year 2010 or prior.

Response: The rationale has been adjusted and updated with more recent references (line 82-102). The reference list has been updated accordingly.

Once again, we would like to thank the editor and reviewer for their insightful comments and suggestions. We look forward to hearing back from you soon.